# Drought Protective Effects of Exogenous ABA and Kinetin on Lettuce: Sugar Content, Antioxidant Enzyme Activity, and Productivity

**DOI:** 10.3390/plants13121641

**Published:** 2024-06-14

**Authors:** Martynas Urbutis, Irina I. Vaseva, Lyudmila Simova-Stoilova, Dessislava Todorova, Audrius Pukalskas, Giedrė Samuolienė

**Affiliations:** 1Lithuanian Research Centre for Agriculture and Forestry, Institute of Horticulture, Kauno Str. 30, LT-54333 Kaunas, Lithuania; 2Institute of Plant Physiology and Genetics, Bulgarian Academy of Sciences, Acad. Georgi Bonchev Str., Block 21, 1113 Sofia, Bulgariadessita@bio21.bas.bg (D.T.)

**Keywords:** *Lactuca sativa*, exogenous phytohormones, antioxidant, sugar metabolism, productivity

## Abstract

Drought is an environmental stressor that significantly impacts plant growth and development. Comprehending the complexity of drought stress and water utilization in the context of plant growth and development holds significant importance for sustainable agriculture. The aim of this study was to evaluate the effect of exogenously applied phytohormones on lettuce (*Lactuca sativa* L.) sugar content profiles and antioxidant enzyme activity and productivity. Lettuce plants were grown under normal and drought conditions in a growth chamber with a photoperiod of 14/10 h (day/night). Kinetin and abscisic acid were applied separately and in combinations when the second leaf was fully expanded. The results showed that sugar accumulation and productivity of the pretreated plants under drought were significantly higher than the controls. The perspective offered by this work showed that growth-related and stress-related phytohormones significantly influenced plant sugar metabolism, metabolic profiles, and productivity, thus enabling the control of yield and quality.

## 1. Introduction

Drought is a significant environmental stressor that impacts the growth and development of plants [1]. Comprehending the complexity of drought stress and water utilization in the context of plant growth and development holds significant importance for sustainable agriculture. This study was conducted in order to understand phytohormone impact on plant sugar metabolism, antioxidant enzyme activity, and productivity. Understanding these processes offers an idea of how plants react to environmental stressors and offers strategies to preserve yield under adverse conditions.

Cytokinins (CKs) and abscisic acid (ABA) play important roles in plant development and drought stress response [2]. Cytokinins influence plant growth by inducing alterations in plant morphology and metabolism [3,4]. They activate various physiological reactions, enhancing the likelihood of plant survival under drought [5]. Evidence supports the significant role of CKs in the growth and development of roots across various plant species. Previous studies suggest that root formation is shaped by sugar metabolism, as well as the signaling pathways involving auxin and CKs [6]. Drought stress has an impact on photosynthesis, primarily by causing stomatal closure and metabolic dysfunction [7]. Cytokinins, influencing various aspects of the photosynthetic machinery, can counteract the reduction in photosynthetic rate induced by water deficit. This compensation occurs through the regulation of stomatal conductance or chlorophyll biosynthesis by CKs [8]. Earlier research supports the idea that increased endogenous CK levels can enhance plant photosynthetic rates during drought. The protection against the degradation of photosynthetic machinery may be attributed to the activation of BR-associated pathways, positively influenced by increased CK levels in IPT-transgenic tobacco plants [9]. Recent studies have indicated that BRs promote CO_2_ assimilation, the quantum yield of photosystem II (PSII), and PSII protection in herbicide-treated plants [10]. Additionally, CKs can stimulate the production of photosynthetically active pigments involved in the light-dependent phase of photosynthesis and the key enzymes of the light-independent phase [11].

Multiple studies have shown that the exogenous application of ABA triggers the activation of the antioxidant defense system and that it is capable of improving plants under unfavorable environmental conditions [12,13,14]. It has been concluded that exogenous ABA has a positive effect on the expression of genes encoding enzymes from the ascorbate-glutathione cycle in experiments with maize grown under water deficit [15]. Besides being involved in the regulation of the antioxidant system, transcriptomic analyses have pointed out the important role of ABA in flavonoids synthesis and its positive effect on the transcription of genes involved in pigments metabolism, including carotenoids biosynthesis [16].

Previously conducted research indicates that during periods of drought stress, the plant hormone abscisic acid (ABA) is produced in the roots and transported to the leaves through xylem vessels. This results in an increased ABA concentration in the leaves [17]. Drought stress activates potassium (K^+^) efflux while suppressing K^+^ influx channels in the membrane of leaf cells. Furthermore, turgor pressure is regulated by activating channels for K^+^, calcium (Ca^2+^), and anions, influencing the movement of ions in and out of cells. This process, induced by ABA, ultimately leads to stomatal closure to minimize water loss [18]. There is a close relationship between the levels of endogenous ABA and the initiation of leaf senescence. Increased ABA content raises the concentration of Ca^2+^ in cells, contributing to leaf senescence [19,20].

Phytohormones are often included in the composition of biostimulants, as they are capable of ameliorating plant stress responses. Cytokinins are the main compounds in plant growth and development during the vegetative stage. Their effects as priming agents improve seed germination and seedling strength. Cytokinins have been linked to enhanced cell division rates [21], and they downregulate the expression of *ABI5* (*ABA INSENSITIVE*), which counteracts the inhibitory role of ABA [22]. Kinetin (KIN) is a synthetic cytokinin with growth-promoting properties. It is capable of inducing cell division and differentiation. Applied exogenously, it stimulates chlorophyll synthesis and nutrient mobilization [23], and it strengthens plants under drought conditions [24,25]. Previously, the stress-improving effects of exogenous kinetin through the activation of antioxidative non-enzymatic and enzymatic defenses were demonstrated [26,27,28]. In a comparative transcriptomic analysis of *Arabidopsis thaliana* and tobacco, the unique property of kinetin to activate the flavonoid synthesis genes was demonstrated [29]. Another study revealed protective KIN role through the modulation of cross-talk between various phytohormones involved in stress responses [30].

These aspects of ABA and kinetin action offer a credible base to test their protective properties under unfavorable growth conditions, including drought stress. In our study, we evaluated different concentrations of exogenous applications of kinetin and abscisic acid, administrated separately or in combination. The effects of the phytohormones on antioxidant enzyme activity and sugar metabolism were evaluated under normal conditions and under moderate drought. The experimental group that showed the best performance under drought stress was subjected to profiling of the antioxidant enzyme activities via RT-qPCR and in-gel staining enzyme activity analyses.

## 2. Results

The exogenous phytohormone impact on growth and metabolite profiles of lettuce plants grown under normal conditions and moderate drought stress was evaluated.

Compared to the control, under normal growth conditions, the application of K20 (kinetin 20 mg L^−1^) led to a significant decrease in fructose content, showing a reduction of 46% (Figure 1). Similarly, A20K40 (abscisic acid 20 mg L^−1^ + kinetin 40 mg L^−1^) resulted in a reduction of 21%, while A30K30 (abscisic acid 30 mg L^−1^ + kinetin 30 mg L^−1^) increased the fructose amount by 34%.

Under drought conditions, different responses were observed. A20 (abscisic acid 20 mg L^−1^) and A40 (abscisic acid 40 mg L^−1^) demonstrated significant increases of 42% and 51% in fructose content. Furthermore, A30K30 significantly increased the fructose amount by 56%, compared to the control.

Under normal growth conditions, the application of K20 resulted in a significantly decreased glucose content per weight of 52% (Figure 2). Furthermore, under the same conditions, A20K40 exhibited a reduction of 26%, while A30K30 demonstrated an increase of 24% in glucose amount.

Under drought conditions, different responses in glucose content were observed (Figure 2). A20 demonstrated a significant increase in glucose (48%), and A40 displayed a notable rise of 51% in comparison to the control (plants grown under normal conditions). Furthermore, the response of A30K30 showed a significant increase of 54% in glucose content.

Under normal growth conditions, individual phytohormone treatments exhibited significant reductions in sucrose content, compared to the control (Figure 3). Specifically, K20 demonstrated a reduction of 68%, K40 (kinetin 40 mg^−1^) was reduced by 43%, A20 showed a reduction of 51%, and A40 (abscisic acid 40 mg L^−1^) exhibited a decrease of 53% (Figure 3). Furthermore, when examining phytohormone mixtures, significant reductions in sucrose content of the well-watered experimental groups were observed in A30K30 (by 27%) and A40K20 (abscisic acid 40 mg L^−1^ + kinetin 20 mg L^−1^) (by 30%)compared to the control that did not receive any hormonal treatments.

Under drought conditions, both individual and combined treatments demonstrated increased responses of sucrose accumulation (Figure 3). K20 and K40 were significantly increased by 96% and 82% in comparison to the control. Notably, the treatment with abscisic acid (A40) led to a significant increase in sucrose content by 225%.

Under normal growth conditions, no significant differences in maltose content were observed (Figure 4), except in the group treated with A40 in which the levels of maltose were under the detection limits of the used method.

However, under drought conditions, distinct responses were observed. Individual phytohormone treatments demonstrated a significant increase in maltose content. Specifically, K20 increased by 40%, and K40 increased by 36%, compared to the control. Moreover, plants treated with abscisic acid displayed significant maltose content increases: A20D increased by 42% and A40D by 47%. Noteworthy increases were also observed with phytohormone mixtures under drought conditions. A20K40 (abscisic acid 20 mg L^−1^ + kinetin 40 mg L^−1^) showed a substantial increase of 73%, and A30K30 demonstrated the most significant rise, reaching 140%, compared to the control, while A40K20 exhibited an increase of 44%.

Under normal growth conditions, the K40 treatment group exhibited a significant reduction in fresh weight, showing a 38% decrease, compared to the control (Table 1). Moreover, regarding the accumulation of dry matter, under normal conditions, variants K20 and K40 showed significant reductions of 39% and 41%, respectively, compared to the control. Under drought conditions, K20D and K40D displayed even more pronounced reductions, being 46% and 39% significantly lower, compared to the control. However, during drought conditions, various treatments revealed significant decrease in biomass accumulation.

It is worth emphasizing that although under normal conditions, there was a slight but significant reduction in almost all of the treatment groups, under a limited water supply, the plants pretreated with the two hormones (either alone or in combination) tended to have a larger leaf area, compared to the non-treated drought-stressed plants.

The antioxidant enzyme activity usually increases under stress to prevent the accumulation of harmful reactive oxidative species (ROS). Comparative analyses of the separate and the combined treatment with ABA and KIN on the enzymes from the antioxidative defense system were evaluated under control and moderate drought conditions. We monitored the gene expression profiles and the enzyme activity in the group that was pretreated with equal concentrations of the two hormones containing 30 mg/L ABA and 30 mg/L KIN (A30K30), as in general, it was the best performing one under drought stress.

Superoxide dismutase (SOD, EC 1.15.1.11) catalyzes the reaction in which the extremely harmful superoxide anion (O_2_^−^) is neutralized by converting it into hydrogen peroxide, water, and oxygen (Figure 5). There are several isoforms, each with a unique subcellular location: Cu/Zn-SODs are active in the cytosol, the extracellular space, and the chloroplast; Fe-SOD isoforms are found in the chloroplast; and Mn-SODs function in the mitochondrion and peroxisomes [31]. The total SOD activity measured by in-gel staining assay showed certain fluctuations, which were not statistically significant, but the treatment with the hormonal mix showed a slight reduction upon dehydration, compared to the respective control. The inspection of the signals linked to the different SOD isoenzymes, however, showed that the applied stress inhibited Mn-SOD in most of the drought-stressed experimental groups, except in the samples subjected to the combined hormonal treatment (Figure 5a).

The RT-qPCR analyses of three *Lactuca sativa* SOD genes (*Ls Mn-SOD*, *Fe-SOD*, and *Cu/Zn-SOD*) showed that ABA applied separately and in combination with KIN induced the accumulation of *Mn-SOD* transcripts under dehydration (Figure 5b). The combined ABA/KIN treatment also upregulated *Fe-SOD* expression in dehydrated plants. The fluctuation of *Cu/Zn-SOD* transcript levels as a result of the exogenous hormonal application in the different experimental groups was not statistically significant.

Catalase (CAT), which operates mainly in peroxisomes, detoxifies excessive amounts of hydrogen peroxide that accumulates under stress and converts H_2_O_2_ into water and oxygen (Figure 6). The in-gel activity staining of catalase (EC 1.11.1.6) revealed a distinct pattern comprising two separate signals in the controls and an additional band, which appeared in the drought-stressed samples (Figure 6a). CAT enzyme profiling revealed relatively stable activity in both the control and the drought-stressed samples derived from hormonal mix-treated plants. The other experimental groups (mock-, ABA- and KIN-treated) consistently exhibited an increase in catalase activity under dehydration (Figure 6a).

The RT-qPCR profiling of the *L. sativa CAT* gene (GeneBank ID: LOC111878432) revealed that the changes provoked by ABA and KIN treatments were not statistically significant, both in the controls and in the plants subjected to drought. However, when applied in combination, the two hormones tended to stabilize the *CAT* gene transcript levels (Figure 6b).

Peroxidases (POX, EC 1.11.1.7) comprise a big family of isoenzymes that oxidize aromatic electron donors (marked with “A” on the figure), including pyrogallol and guaiacol, at the expense of H_2_O_2_ (Figure 7a). They operate in the cytosol, cell walls, and vacuoles. Peroxidase in-gel staining showed a significant increase in the enzyme activity only in the ABA-primed plants subjected to drought. The isoenzyme profiles revealed that this aligned with the enhanced POX3 activity staining signal (Figure 7a).

The expression of the *POX N1* gene coding a secretory metalloenzyme (horseradish-type peroxidase) was significantly upregulated in both ABA- and KIN-treated plants grown under dehydration (Figure 7b). However, this effect was nullified in the drought-stressed group subjected to the combined hormonal pretreatment. Although less pronounced, a similar trend was observed in the expression profiles of the other tested peroxidase genes *POX5* and *APX2*. The increase in the *APX2* transcripts coding an ascorbate-oxidizing cytosolic enzyme showed a statistically significant difference, compared to the control group (Figure 7b).

## 3. Discussion

The presented study focuses on the effects of exogenously applied kinetin and abscisic acid on sugar accumulation and antioxidant enzyme activity in lettuce.

The obtained results on fructose accumulation in lettuce align with the findings of other researchers regarding the pivotal role of sugars, particularly sucrose, glucose, and fructose, in plant growth and development [32]. The results confirm previous findings on the positive effect of exogenous phytohormones in modulating fructose levels in lettuce [32]. Furthermore, the observation that sucrose content peaks during the early stage of stem development in lettuce, storing energy for specific growth stages, is in line with our results demonstrating the influence of exogenous factors on sugar accumulation.

Moreover, the conversion of sucrose into glucose and fructose during later growth stages, as discussed by other authors, aligns with our findings of increased glucose and fructose levels alongside decreased sucrose content [33]. This conversion mechanism likely underlies the dynamic changes in sugar composition observed in response to diverse growth conditions, which are probably regulated by different hormonal signals [33,34]. The significant fructose accumulation resulting from abscisic acid treatments suggests the potential hormonal regulation of sugar metabolism.

Our findings are consistent with the previously published results by Dörffling et al. [35], who showed that accumulations of glucose and sucrose are influenced by exogenous KIN and ABA. In their study, Zhou et al. [36] highlighted that the plants pretreated with phytohormones and exposed to a low nitrogen supply tended to maintain a physiological sucrose level [36]. However, under drought conditions, a different trend was observed, and a significant increase in sucrose content was registered after KIN and ABA applications.

Maltose plays a role in energy metabolism, stress response, and maintaining cellular homeostasis, all of which are vital for plant growth and development [37]. Our experimental data showed that in general, exogenous applications of kinetin and abscisic acid did not result in increased maltose levels under normal growth conditions. However, the observed increase in maltose in the pretreated plants under drought corresponded to previously published data, outlining the important role of this carbohydrate in maintaining plant metabolism under osmotic stress in particular [38].

Previously, it was demonstrated [39] that under salt stress, exogenous abscisic acid application can enhance growth and antioxidant enzyme activity in lettuce seedlings. In our experimental model, we observed the same effect. ABA-pretreated *L. sativa* plants showed increased antioxidant enzyme production when subjected to drought stress. Particularly convincing was the documented upregulation of peroxidase activity in the ABA-pretreated samples that experienced dehydration. This was apparent in both protein and gene expression levels, supporting the relevance of strategies implementing exogenous ABA as a priming agent for the activation of plant defense mechanisms. It should be noted that regardless of the positive effect of KIN on POX gene expression, a synergistic effect of the two hormones (ABA and KIN) was not observed. On the contrary, in the samples subjected to the combined ABA/KIN treatment, the POX enzyme activation was no longer apparent. This suggests antagonist effects of the two hormones in the control of some peroxidase isoenzymes. The treatment with 30 mg/L kinetin did not show any other significant effects on the monitored antioxidant enzymes. It was demonstrated that ABA-treated Arabidopsis plants experienced lower oxidative damage and that this was linked to the activation of Mn-SOD and Fe-SOD isoenzymes [23]. Our results on the *SOD* gene expression profiles are in line with these previously published observations, as we also documented an ABA-stimulated accumulation of *Mn-SOD* and *Fe-SOD* gene transcripts. The ABA effect was further enhanced in the samples subjected to the combined application with kinetin. However, this trend was not found to be statistically significant in the SOD in-gel activity staining profiles. Catalase gene expression and enzyme activity were not influenced by the application of ABA either.

The larger leaf area and higher fresh weight of the hormone-treated plants under drought conditions demonstrates that exogenous KIN and ABA could improve drought resistance. Previous studies on lettuce grown under stress conditions also highlight the complex role of sugars in plant growth and development [13,32,37]. However, the beneficial impact of exogenous phytohormones on lettuce physiology needs to be validated under field conditions.

## 4. Materials and Methods

### 4.1. Growth Conditions

Lettuce plants (*Lactuca sativa* L. cv. Lobjoits; Green Cos, CN seeds Ltd., Cambridgeshire, UK) were grown on 44 mm Jiffy^®^ Peat Pellets (Jiffy Growing Solutions, Zwijndrecht, The Netherlands) saturated with water at 80% of the substrate’s field capacity (FC). The plants were kept in a growth chamber with an air humidity of 60%; an illumination of 200 μmol m^−2^ s^−1^ photon flux density was provided by fluorescent lamps, with a 21/18 °C day/night temperature and a 14/10 h photoperiod. On day 14, when the seedlings had a fully expanded second leaf, some of the plants were sprayed with different concentrations of abscisic acid (ABA) and kinetin (KIN), applied alone or in combination. The following treatments were performed: a control group (sprayed with dH_2_O); KIN 20 mg L^−1^; KIN 40 mg L^−1^; ABA 20 mg L^−1^; ABA 40 mg L^−1^; MIX 1—KIN 20 mg L^−1^ + ABA 40 mg L^−1^; MIX 2—KIN 30 mg L^−1^ + ABA 30 mg L^−1^; and MIX 3—KIN 20 mg L^−1^ + ABA 40 mg L^−1^. Each solution contained 100 µL of 80% polyether modified siloxane (Ikarai, Kėdainiai, Lithuania), which is a non-toxic, environmentally friendly surfactant, enabling better absorption. The amount of solution applied was 50 mL per 0.25 m^2^. The phytohormones were obtained from Carl Roth (Karlsruhe, Germany), Sigma-Aldrich (Darmstadt, Germany). Part of the differently treated plants were subjected to gradual dehydration for 5 days until the substrate humidity reached 30% field capacity (FC). It was then maintained for an additional 5 days. The plant material was freeze-dried or frozen in liquid nitrogen and stored at −70 °C until analyzed.

### 4.2. Sample Preparation for Determination of Sugars by HPLC

An amount of 0.05 g of freeze-dried plant material was weighed in 5 mL plastic microtubes (Th. Geyer GmbH & Co. KG, Renningen, Germany) mixed with 2 mL of warm (40–50 °C) deionized water and then was rotated for 2 h on the LABINCO LD79 digital test tube rotator (Labinco BV, Breda, The Netherlands). After mixing, the tubes were centrifuged at 4500 rpm in a Z366K centrifuge (HERMLE Labortechnik GmbH, Wehingen, Germany) for 15 min. After centrifugation, 0.9 mL of supernatant was transferred into a 2 mL microtube and mixed with 0.9 mL of 0.01% ammonium acetate solution in acetonitrile. The obtained solution was stored in a refrigerator at 4 °C for 30 min. After that, microtubes with samples were centrifuged at 14,000 rpm for 15 min in a MiniSpin centrifuge (Eppendorf AG, Hamburg, Germany). After centrifugation, samples were filtered through 0.22 µm nylon syringe filters into chromatography vials.

### 4.3. Determination of Sugars by HPLC

Contents of saccharides were evaluated using a Shimadzu HPLC system (Shimadzu, Kyoto, Japan) consisting of a DGU-20A5 vacuum degasser, LC-30AD HPLC pumps, a SIL-30AC autosampler, a CTO-20AC column oven, an ELSD-LT II light scattering detector, and a CBM-20A communication bus module. Separation of compounds was performed using gradient elution with ultrapure water (solvent A) and acetonitrile (Solvent B) on a Shodex HILICpak VG-50 4D column (150 × 4.6 mm, 5 µm) (Shova Denko Europe GmbH, Germany) at 35 °C; a flow rate of 0.8 mL/min was used. The gradient was formed as follows: Initially, 77% of solvent B was used; then, in 11.5 min, the concentration of B was decreased to 71%. In the following 1.5 min, the gradient was returned to initial conditions and held for 2 min. A sample injection volume of 10 µL was used. Quantitation of compounds was performed in the ELSD chromatogram. The external calibration standard method (standard concentration ranged from 0.025 to 1 mg/mL) was used for the quantitation of individual saccharides.

### 4.4. Non-Denaturing Electrophoresis and In-Gel Activity Staining of Antioxidant Enzymes

The in-gel activity staining of peroxidase (POX), catalase (CAT), and superoxide dismutase (SOD) activities were performed as previously described by Vaseva et al. [40]. Total protein extracts (60 µg per lane) were separated on a native 7% PAGE at 4 °C for the determination of catalase and peroxidase activities. Benzidine was used as a substrate in the POX assay. SOD activity staining was performed on 10% native PAGE with 20 µg of total protein load per lane, and the different enzyme types were assigned according to Yildiztugay [41]. The staining intensities of the activity bands were determined with ImageJ 1.52i.

### 4.5. RT-PCR Analysis

The GeneJET Plant RNA Purification Kit (Thermo Scientific, Waltham, MA, USA) was used to extract total RNA from the youngest fully expanded leaf of plants from the different experimental groups: plant grown under normal conditions, plants grown under normal conditions and treated with kinetin (30 mg/L) and ABA (30 mg/L) or their combination, plants subjected to drought, and individuals that did not receive exogenous phytohormones and were drought-stressed. The synthesis of cDNA was performed with 100 ng of total RNA using the RevertAid First Strand cDNA Synthesis Kit (Thermo Scientific™, Vilnius, Lithuania) according to the manufacturer’s instructions. The gene transcript abundance was evaluated by quantitative real-time RT-PCR (qRT-PCR) using 2X GreenMasterMix No ROX^TM^ (GENAXXON Bioscience, Ulm, Germany) with the ‘PikoReal’ Real-Time PCR System (Thermo Scientific, Basel, Switzerland). The PCR program settings were 95 °C for 15 min and 45 cycles of 95 °C for 15 s, followed by 60 °C for 30 s, and melting curve analysis with a temperature range of 60–95 °C in 0.2 °C increments for 60 s. The relative expression of the target genes was calculated by the ΔΔCq method of Livak and Schmittgen [42], using actin (Gene Bank ID: XM_023878805) and 18S ribosomal RNA (Gene Bank ID: AH001680) as references. *Lactuca sativa* genes coding for the analyzed antioxidant enzymes and the respective primers used are listed in Table 2.

### 4.6. Biometric Measurements

Five representative lettuce plants were selected for leaf area measurement (cm^2^), which was performed with the Rosette Tracker plugin in Image J [43]. The dry weight (DW) was determined by drying the plants at +70 °C for 48 h to a constant weight.

### 4.7. Statistical Analysis

Statistical analysis was performed using Microsoft Excel 2016 and Addinsoft XLSTAT 2022.1 statistical and data analyses (Long Island, NY, USA). The data in Figure 1, Figure 2, Figure 3 and Figure 4 and Table 1 are presented as the means of five replicates (n = 5) linked to the sampling points. One-way analysis of variance (ANOVA), followed by Duncan’s significant difference test (*p* < 0.05) for multiple comparisons, was used to evaluate differences between means of measurement. The error bars in the graphs reflect the standard error (SE). RT-qPCR and in-gel activity staining data (Figure 5, Figure 6 and Figure 7) were derived from three biological replicates per experimental group, and the statistical significance was assessed by Welch’s *t*-test.

## 5. Conclusions

This study showed the complex response mechanisms of plants to unfavorable environments. Findings revealed that the accumulation of sugars, specifically glucose, fructose, and sucrose, indicate a metabolic shift of soluble sugars counteracting drought. A30K30 increased fructose, glucose, and maltose metabolism under limited water availability. Furthermore, exogenous phytohormones significantly increased the plant’s leaf area under water deficit. Collectively, our findings present some additional elements of the plant responses to dehydration. The results could be useful in developing new strategies for mitigating drought stress outcomes in vegetable crops.

## Figures and Tables

**Figure 1 plants-13-01641-f001:**
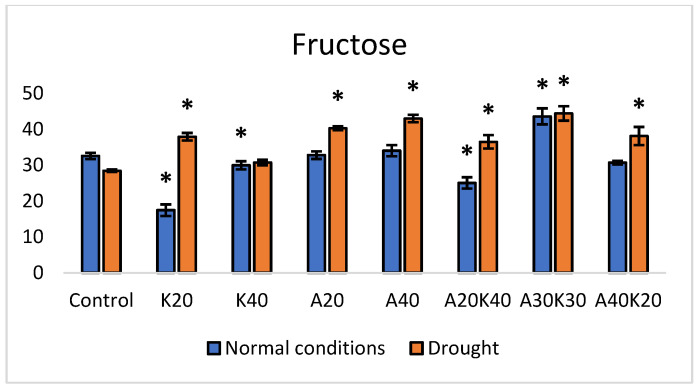
Fructose content (mg/g^−1^ DW) under normal and drought conditions in lettuce treated with different hormone concentrations. Values are mean ± SE of 5 replicates, and * shows significant differences by the Duncan comparisons test (*p*  ≤  0.05). Spraying: Control—water, K—kinetin, A—abscisic acid.

**Figure 2 plants-13-01641-f002:**
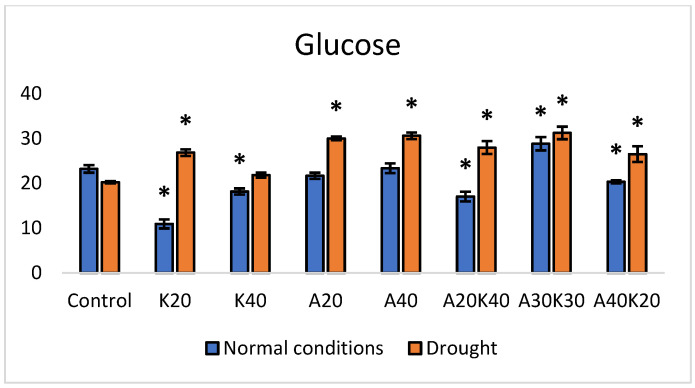
Glucose content (mg/g^−1^ DW) under normal and drought conditions in lettuce treated with different hormone concentrations. Values are mean ± SE of 5 replicates, and * shows significant differences by the Duncan comparisons test (*p*  ≤  0.05). Spraying: Control—water, K—kinetin, A—abscisic acid.

**Figure 3 plants-13-01641-f003:**
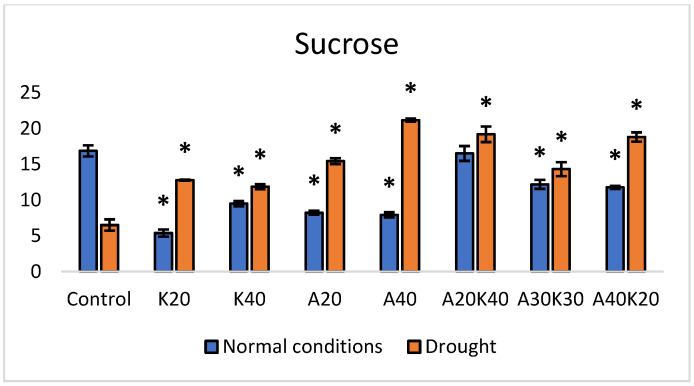
Sucrose content (mg/g^−1^ DW) under normal and drought conditions in lettuce treated with different hormone concentrations. Values are mean ± SE of 5 replicates, and * shows significant differences by the Duncan comparisons test (*p * ≤  0.05). Spraying: Control—water, K—kinetin, A—abscisic acid.

**Figure 4 plants-13-01641-f004:**
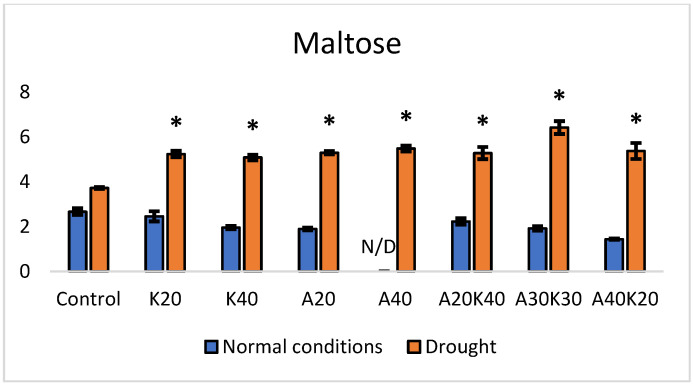
Maltose content (mg/g^−1^ DW) under normal and drought conditions in lettuce treated with different hormone concentrations. Values are mean ± SE of 5 replicates, and * shows significant differences by the Duncan comparisons test (*p*  ≤  0.05). Spraying: Control—water, K—kinetin, A—abscisic acid, “N/D” indicates “not detected”.

**Figure 5 plants-13-01641-f005:**
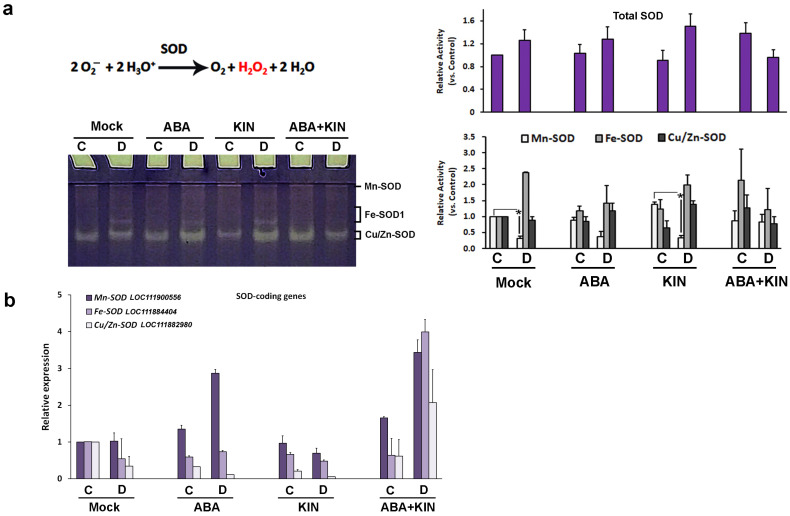
SOD in-gel activity staining (**a**) and transcript profiling of *Lactuca sativa* SOD-coding genes (**b**) in plants grown under normal or drought-stressed conditions and pretreated with ABA (30 mg/L), KIN (30 mg/L), or their combination. Representative native gel visualizes the SOD activity staining assay. The bar charts are based on three independent repetitions and represent the quantification of the total SOD, the measured signal strengths of the Mn-SOD, Fe-SOD, and Cu/Zn-SOD isoenzymes (**a**), and *SOD* genes transcript profiling (**b**). The asterisks mark significant differences between the controls in each treatment group (n = 3, *p* ≤ 0.05, Welch *t*-test).

**Figure 6 plants-13-01641-f006:**
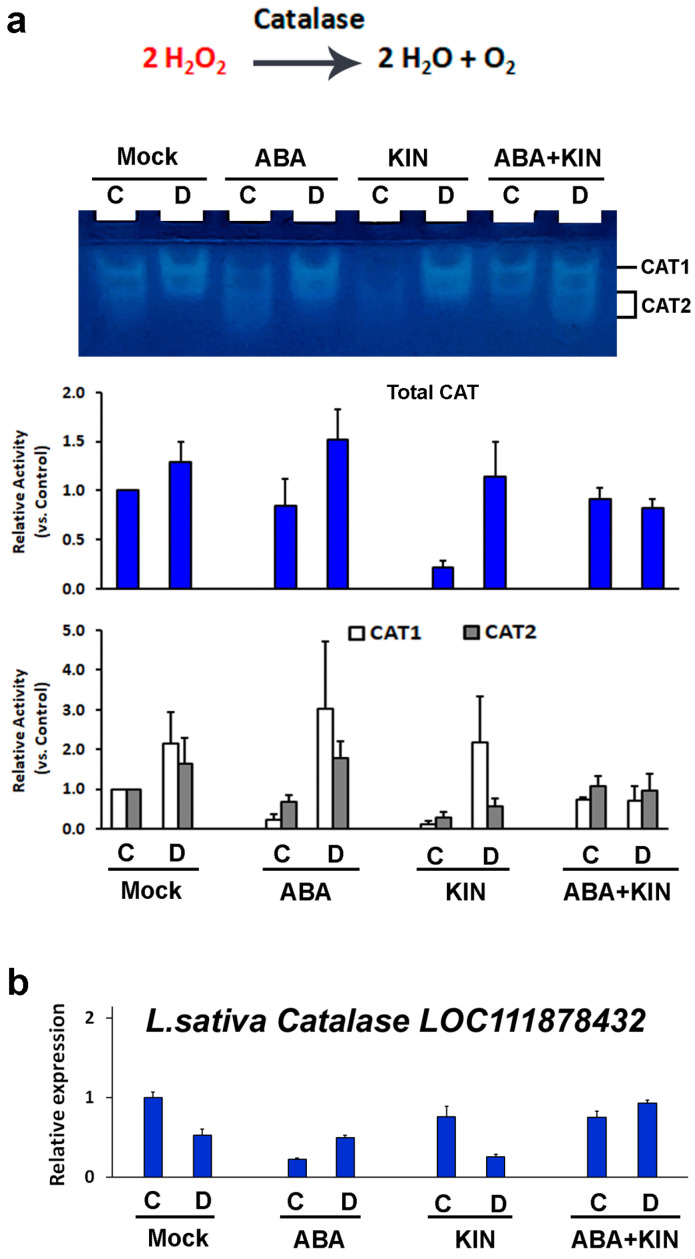
CAT in-gel activity staining (**a**) and transcript profiling of *Lactuca sativa CAT* gene (**b**) in plants grown under normal or drought-stressed conditions and pretreated with ABA (30 mg/L), KIN (30 mg/L), or their combination. Representative native gel visualizes the CAT activity staining assay. The bar charts are based on three independent repetitions and represent the quantification of the total CAT activity staining and the measured signal strengths of the identified CAT isoenzymes (**a**) and the *CAT* transcript profiling (**b**).

**Figure 7 plants-13-01641-f007:**
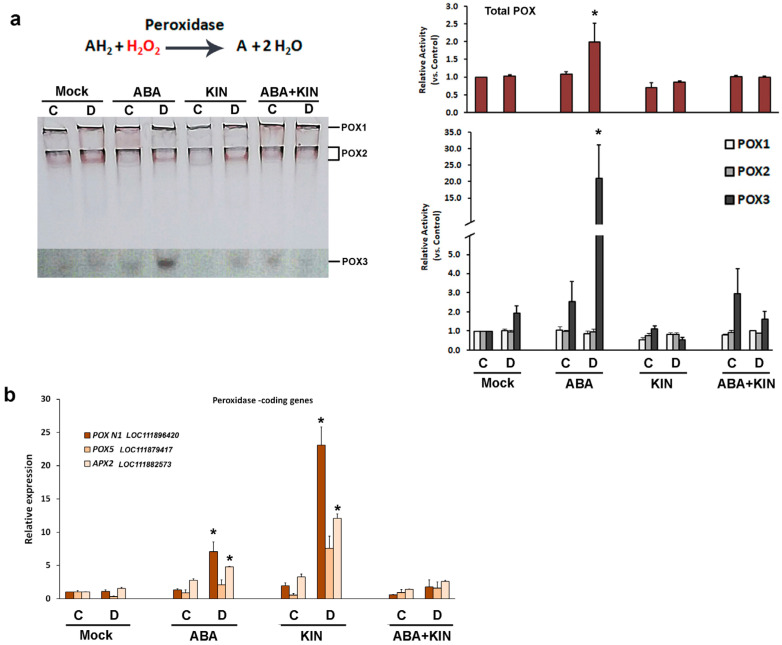
POX in-gel activity staining (**a**), and transcript profiling of *Lactuca sativa* peroxidase-coding genes *POX N1*, *POX5*, and *APX2* (**b**) in plants grown under normal or drought-stressed conditions and pretreated with ABA (30 mg/L), KIN (30 mg/L), or their combination. The representative native gel of the POX activity staining assay is shown. The contrast of the lower sector of the gel was enhanced for better visualization of the POX3 signal. The bar charts are based on three independent repeats, and they represent the quantification of the total peroxidase activity and the measured signal strength of the identified POX isoenzymes (**a**) and the *POX* genes transcript profiling. The asterisks mark significant differences between the respective controls in each treatment group (n = 3, *p* ≤ 0.05, Welch *t*-test).

**Table 1 plants-13-01641-t001:** Effects of phytohormones on lettuce growth. Values are mean ± SE of 5 replicates, and * shows significant differences by the Duncan comparisons test (*p*  ≤  0.05). Spraying: Control—water, K—kinetin, A—abscisic acid.

	Fresh Weight [g]		Dry Weight [g]		Leaf Area [mm^2^]	
	Normal Conditions	Drought	Normal Conditions	Drought	Normal Conditions	Drought
Control	2.50	1.43	0.16	0.11	6237	2828
K20	1.67	1.20 *	0.10 *	0.09 *	4844 *	3662 *
K40	1.54 *	1.28 *	0.09 *	0.10	4885 *	3301 *
A20	1.86	1.31 *	0.11	0.10	5740 *	2806 *
A40	2.40	1.42	0.13	0.10	5612 *	3561 *
A20K40	1.90	1.27 *	0.10	0.09 *	5778 *	3490 *
A30K30	1.96	1.49 *	0.11	0.11	5002 *	3558 *
A40K20	2.09	1.36 *	0.11	0.10	6001	3600 *

**Table 2 plants-13-01641-t002:** Analyzed *Lactuca sitiva* genes coding for antioxidant enzymes and primer pairs used in the qRT-PCR.

Gene Name	Locus	Forward Primer (5′-3′)	Reverse Primer (5′-3′)
*Ls18S RNA*	AH001680	CGGGTGACGGAGAATTAGGG	TACCTCCCCGTGTCAGGATT
*Ls Actin-7*	LOC111882438	CTGGTGATGGTGTCTCCCAC	GGCGAGCTTCTCCTTCATGT
*Ls Mn-SOD*	LOC111900556	CACCCCAGTATTTGGATGGCT	CTCCCTCCCCCTATGTGCTA
*Ls Fe-SOD*	LOC111884404	GGGAATCCATGCAACCAGGA	AAAACAAGCCAAACCCAGCC
*Ls Cu/Zn-SOD*	LOC111882980	CACTCTTACAGACGCTTTGCG	ATGGTGCCACTAACACCCTC
*Ls Catalase*	LOC111878432	GCCATGCTGAACAGTACCCT	TCTCTCTCCTGGCTGCTTGA
*Ls APX2*	LOC111882573	GACATCGGCGATCTTCTGGT	TCTCGAAGCTTCCTCTTCGC
*Ls POX N1*	LOC111896420	CTATGGTTGATATTGGCGTCGT	ACAAAGTCGGCCATTGGAGAT
*Ls POX5*	LOC111879417	GGTCGCTAAAGCCTACTCCC	ACTTGGGTTGTTTGCTGGTG

## Data Availability

Data are contained within the article.

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
