# Peer review of "Drought Protective Effects of Exogenous ABA and Kinetin on Lettuce: Sugar Content, Antioxidant Enzyme Activity, and Productivity"

_plants, 2024, doi:10.3390/plants13121641_

Round 1
Reviewer 1 Report
Comments and Suggestions for Authors
Manuscript : Drought protective effects of exogenous ABA and kinetin on 2 lettuce: sugar metabolism, metabolic profiles and antioxidant enzyme activity, and productivity by Martynas Urbutis, Irina Vaseva , Liyudmila Simova-Stoilova , Dessislava Todorova , Audrius Pukalskas and GiedrÄ— SamuolienÄ— is interesting. However, in my opinion, this paper need to be improved.
Intoroduction
-
The purpose of the study should be emphasized. What is the importance of the research described: biological-physiological, economic, etc
-
please explain why lettuce was used as the research model
Materials and methods
-
Please explain whether the method described in point 4.3. Is it a method developed independently or taken from the literature? If from the literature, please add a reference to the method described
-
Complete data in line 390
Disussion
-
Please explain in more detail the mechanism of the influence of exogenous kinetin and ABA on sugar levels and the importance of this process.
-
Line 260 - please complete the literature reference
-
Paragraph between lines 269 and 278 - please complete the literature reference
-
What is the importance of maltose in the physiological processes of plants and why did the authors decide to examine its level under the influence of kinetin and ABA. Complete the discussion, please
-
Have other authors examined the effect of ABA on sugar levels after stressors? What is the impact of ABA and kinetin on sugar metabolism and the level of antioxidant. enzymes after stressors other than drought? Complete the discussion, please
-
What is the mechanism of the observations described in the last paragraph of the discussion
Author Response
Dear Reviewer,
We would like to express our sincere gratitude for taking the time to review our manuscript and for providing thoughtful and insightful comments. Your feedback has a great value in improving the quality and clarity of our paperwork. We made adjustments accordingly. Please, check attached file for our comments.

Reviewer 2 Report
Comments and Suggestions for Authors
The manuscript "Drought protective effects of exogenous ABA and kinetin on 2 lettuce: sugar metabolism, metabolic profiles and antioxidant 3 enzyme activity, and productivity" looked at the effect of kinetin and ABA on lettuce growth and physiology. The manuscript is well written. The major concern i have on the table for FW and DW where it seems growth hormone decreased the biomass whereas conclusion and discussion says opposite. Further authors need to specify the design of experiment used. For specific comments please refer to attached.

Author Response
The manuscript "Drought protective effects of exogenous ABA and kinetin on 2 lettuce: sugar metabolism, metabolic profiles and antioxidant 3 enzyme activity, and productivity" looked at the effect of kinetin and ABA on lettuce growth and physiology. The manuscript is well written. The major concern i have on the table for FW and DW where it seems growth hormone decreased the biomass whereas conclusion and discussion says opposite. Further authors need to specify the design of experiment used. For specific comments please refer to attached.
Paper was corrected according to Your suggestions. I also corrected a conclusion.
Reviewer 3 Report
Comments and Suggestions for Authors
Novelty
1. This study investigated the effects of exogenous Kinetin (ABA) and kinetin (Kinetin) on glucose metabolism, metabolic profile, antioxidant enzyme activity and productivity of lettuce under drought conditions, which is of great significance for understanding how plants respond to environmental stress through endogenous hormone regulation.
2. This study used lettuce in a controlled environment as a model plant to systematically evaluate the effects of two plant hormones alone and in combination on plant growth and stress response, which provided a theoretical basis for the future development of novel biostimulants.
3. Through the analysis of antioxidant enzyme activity, the paper provides insights into the potential regulatory mechanisms of hormone treatment on the plant antioxidant defense system, which is novel in the field of plant biology.
4. The thesis aims at drought stress, an important issue in current agricultural production, which makes the research have high practical value and application prospect.
5. Rigorous research methods, including the control of growth conditions, the design of hormone treatment and the determination of biomass and biochemical indicators, ensure the reliability of experimental results.
6. In the discussion part of the paper, the experimental results are analyzed in detail and compared with existing literatures, which shows the author's in-depth understanding of the research field.
Improvement suggestion:
1. Although studies have designed a variety of hormone concentrations and combinations, in-depth discussions on the mechanism of hormone action at the molecular level, such as changes in hormone receptors and activation of signal transduction pathways, are lacking.
2. The paper mainly focused on glucose metabolism and antioxidant enzyme activity, but the influence of other physiological processes such as photosynthesis and water use efficiency was not discussed enough.
3. Although the results show that hormone treatment can improve the drought resistance of plants, the practical application of this effect in field conditions is not fully demonstrated.
4. It is suggested that the authors should add molecular biological methods, such as gene expression analysis and protein interaction studies, in future studies to reveal the molecular mechanism of hormone regulation of drought resistance response.
5. The scope of research may be extended to evaluate the effects of hormone treatment on other important physiological processes of plants, such as photosynthesis and root development, in order to fully understand the physiological basis of hormone regulation.
6. In order to validate the practical application value of laboratory studies, field trials are recommended to evaluate the effect of hormone treatment on improving crop drought resistance in actual agricultural production
Author Response

(The authors gave the same response as above.)

Round 2
Reviewer 1 Report
Comments and Suggestions for Authors
The fragment between lines 305-307 requires correction
Author Response
Dear reviewer,
Thank You one more time for the comments. Correction was made, according to Your suggestions.
Best regards,
Reviewer 3 Report
Comments and Suggestions for Authors
The authors address all the problems. I agree to publish.
Author Response
Dear reviewer,
Thank you for the comments. Best of luck in future work!
Best regards,